# Prediction of Flow Properties of Porous Triply Periodic Minimal Surface (TPMS) Structures

**Saúl Piedra** , **Arturo Gómez-Ortega** and **James Pérez-Barrera** *

CONAHCYT-Centro de Ingeniería y Desarrollo Industrial (CIDESI), Av. Playa Pie de la Cuesta No. 702, Santiago de Querétaro, Querétaro 76125, Mexico; saul.piedra@cidesi.edu.mx (S.P.); arturo.gomez@cidesi.edu.mx (A.G.-O.)
* Correspondence: james_pe_ba@hotmail.com or james.perez@cidesi.edu.mx

**Abstract:** The flow through geometrically complex structures is an important engineering problem. In this work, the laminar flow through Triply Periodic Minimal Surface (TPMS) structures is numerically analyzed using Computational Fluid Dynamics (CFD) simulations. Two different TPMS structures were designed, and their porosity was characterized as a function of the isovalue. Then, CFD simulations were implemented to compute the pressure drop by systematically varying the flow velocity and the porosity of the structure. A Darcy–Forchheimer model was fitted to CFD results to calculate the inertial and permeability coefficients as functions of the porosity. These types of results can be very useful for designing fluid flow applications and devices (for instance, heat exchangers), as well as for integrating these TPMS structures since the flow can be very well estimated when using the porous medium model.

**Keywords:** triply periodic minimal surfaces; Darcy–Forchheimer porous media model; computational fluid dynamics

## 1. Introduction

The flow through porous media is an important engineering problem since many industrial processes need a liquid to flow through (or to be extracted from) natural or man-made porous structures as part of the productive process. These processes pose a challenge since the fluid permeates the interstitial (or pore) space available between the solid matter, thus making it difficult for fluid flow. In addition, this occurs depending on the particular features of the medium. These and other related phenomena have been treated by a combination of experimental and theoretical approaches, therefore resulting in the proposal of several simplified models (currently known as porous media models) that try to take into account the complexity of the medium's geometry through a few characteristic parameters. Porous media models allow for the assessment of the qualitative and quantitative behavior of the flow that moves through geometrically complex structures, and they can be used without the need for detailed and computationally expensive numerical simulations or with experimental observations and measurements that are hard to achieve.

The pioneering model by Darcy considers a linear relation between the macroscopic pressure gradient that occurs through the media and the resulting average velocity of the fluid. In this model, the two characteristic parameters of importance are porosity (understood as the ratio of void space to total volume) and permeability (which is dependent on pore shape, connectivity, and flow features). Although this model has been proven to be valid only for very small Reynolds numbers, its conceptual simplicity has served as the basis for more complete models. One such of these models was proposed by Forchheimer [1], in which a quadratic inertial term was used to modify Darcy's model.

This improved model, known as the Darcy–Forchheimer (DF) model, has been used to describe the fluid flow through porous media in different situations. For instance, Tosco et al. [2] performed 2D pore-scale flow simulations for Newtonian and non-Newtonian

fluids, and they concluded that the DF model can accurately describe the flow behavior for both kinds of fluids when using only two macroscopic parameters for the Newtonian case and three for non-Newtonian fluids. In [3], the authors also studied the flow of a non-Newtonian (viscoelastic) fluid, whereby they considered the Cattaneo–Christov heat transfer through a DF porous medium over a linear stretching surface. In a more recent publication, Hayat et al. [4] analyzed the flow through a DF porous medium over an inclined stretching surface, which included the consideration of mixed convection, heat transfer, and chemical reactions. As demonstrated by latter references, the interest of the scientific community in this kind of system includes the analysis of flows through a DF porous medium alongside different physical phenomena, such as magnetohydrodynamics [5,6], homogeneous–heterogeneous chemical reactions [7], mass transfer [8], and nanofluids [9], to name a few. Some works have even combined several of these phenomena to consider multiphysics in their analysis [10–13]. One interesting application of the DF model corresponds to the analysis and characterization of biological systems [14,15]. Takhanov [16] recorded a detailed literature review of this model and its derivation by using both micro- and macroscale approaches and established a range of validity with experimental data. In their work, the authors stated that this model can be derived at the continuum scale from Navier–Stokes equations, whereas, in a microscopic view, a porous medium is thought as an array of parallel or series of capillary tubes, for which simple analytical solutions can be applied. Regardless of the approximation used, the author assessed the validity of the model using previously published experimental data for consolidated and unconsolidated porous media, as well as for rock fractures.

Additive manufacturing (AM), most commonly known as 3D printing, comprises a set of seven families of different technologies that are used to fabricate prototypes and devices in a layer-by-layer manner. The rapid technological growth of AM provides engineers with almost unlimited freedom of design, as well as a set of advantages over traditional manufacturing techniques, such as the possibility to build lightweight structures [17], topologically optimized parts [18], and part consolidation [19,20], all of which are based on the fact that AM allows for the printing of complex structures easily. In particular, Triply Periodic Minimal Surface (TPMS) structures fabricated by AM are promising complex structures for the enhancement of different applications in which fluid flow is involved. In this sense, the fluid flow through TPMS-based structures has been reported previously, although most of the studies regarding this have focused on the heat transfer properties of these structures [21,22]. A detailed summary of these investigations can be found in [23]. TPMS structures have also raised interest in biological applications [24] and tissue engineering [25].

Recently, several papers have considered TPMS structures as porous media. For instance, Rathore et al. [26] performed pore-scale simulations through four different TPMS structures when using different approaches. They achieved this by considering the TPMS structures in the following ways: (1) as a zero-thickness wall; (2) as the interface between a solid and a fluid; and (3) as a porous medium, which was modeled by an extended Darcy model. Through their simulations, the authors characterized the flow through the four TPMS structures by varying the flow velocity for a single porosity value of 32%. They found that the extended Darcy porous medium model was appropriate for estimating the pressure drop, and this was achieved by considering a cubic behavior for the average velocity. In their work, Ali et al. [27] studied the flow properties of lattice and TPMS structures with a constant porosity of 80%, and they calculated their permeability using Darcy's law. Zeng and Wang [28] proposed a method through which to construct low porosity, TPMS-based isotropic, porous structures, and they compared their structures with a graphite porous material. As a result, the authors proposed an analytical model for the permeability of the structures based on the Hagen–Poiseuille equation and Darcy's law. Zou et al. [29] designed, 3D printed, and experimentally assessed the permeability of several TPMS structures with the constant hydrostatic head technique, as well as by fitting the results to a Darcian model.

Since TPMS structures are geometrically complex, and as highly precise AM technologies are limited to small printing volumes, the use of this kind of structures has been researched to be applied in heat sinks for electronic devices, in which high heat dissipation, large surface areas, and low-pressure drops are desirable. In this sense, Al-ketan et al. [30] designed, simulated, and assessed the thermal performance of TPMS-based heat sinks under forced convection conditions, and they 3D printed some structures to assess their manufacturability. Although the thermal performance increased, there were defects when fabricating the sinks, and these left their study in a proof-of-concept situation. In their work, Passos [31] integrated several TPMS structures into counterflow heat exchangers (HXs) and numerically investigated and compared their performance against a traditional flat plate HX. Li et al. [32] proposed and simulated a heat exchanger design to be used in a supercritical $CO_2$ Brayton cycle using TPMS structures and compared its thermal and fluid flow features against a printed circuit heat exchanger. They concluded that their design promotes turbulent flows and, thus, increases heat transfer by 30–80%. As can be seen, the use of TPMS in HXs is still an ongoing, active branch of research.

As demonstrated, interest in TPMS structures as porous media has been limited to studies involving just a few porosity values and linear porous media models. In this work, we investigated the flow characteristics through geometrically complex structures by systematically varying their porosity using Computational Fluid Dynamics (CFD) simulations in a channel flow. The analysis of the pressure drop due to the presence of the TPMS structures, varying the inlet flow velocity, leads to the conclusion that the flow through the selected structures behaves as a Darcy–Forchheimer porous medium, whose parameters, namely porosity, permeability, and Forchheimer coefficient were characterized numerically. Given that the fluid flow analysis is an important aspect of heat exchanger design (alongside the heat transfer physics), this work contributes by providing a simplified model for calculating the pressure drop along these structures, which can be seamlessly integrated into the conventional heat exchanger design process.

## 2. Design of Triply Periodic Minimal Surface (TPMS) Structures

Among the complex structures that can be 3D printed, TPMS, which are smooth surfaces with zero mean curvature and that separate space into two regions of the same volume [33], have garnered attention for applications in structural mechanics [34–36], heat transfer [37,38], and biomedical devices [39,40]. This attention is due to their mathematical simplicity and highly tailorable geometric properties, which directly affect different effective physical properties.

TPMS are mathematically defined by level-set equations of the form $F(x, y, z) = t$, where $t$ is known as the isovalue. For $t = 0$, the level-set surface separates space into two regions of the same volume. If $t \neq 0$, the surface offsets so that one of the regions becomes larger while the other one becomes smaller in the same amount. Consequently, the isovalue directly influences the solid volume fraction (or, equivalently, the porosity) of the structure. This type of equation defines a skeletal structure, but squaring the equation results in defining a sheet-based structure.

In this paper, we analyze two TPMS sheet-based structures: The Schwarz Primitive and Schoen I-graph Wrapped Package (IWP) structures, which are defined by the following equations:

$$\text{Primitive:} \quad \cos \tilde{x} + \cos \tilde{y} + \cos \tilde{z} = t, \tag{1}$$

$$\text{IWP:} \quad 2(\cos \tilde{x} \cos \tilde{y} + \cos \tilde{y} \cos \tilde{z} + \cos \tilde{x} \cos \tilde{z}) - (\cos 2\tilde{x} + \cos 2\tilde{y} + \cos 2\tilde{z}) = t, \tag{2}$$

where $\tilde{x} = \dfrac{2\pi x}{a}$, $\tilde{y} = \dfrac{2\pi y}{a}$ and $\tilde{z} = \dfrac{2\pi z}{a}$, $a$ being the unit cell size. Figure 1 shows four sheet-based TPMS structures, including the two used in this study.

Despite their evident applications, software for designing and integrating TPMS into usable devices is still scarce. Although there are a few commercial [41] and open-source options [42–45] recently made available, the former tends to be costly, whereas the latter is

limited in some fundamental aspects, such as the necessity to use third-party software to integrate TPMS into arbitrary geometries.

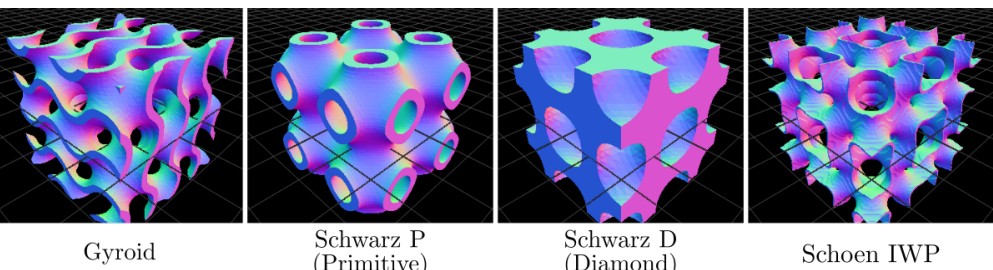

|   |   |   |   |
|---|---|---|---|
| Gyroid | Schwarz P (Primitive) | Schwarz D (Diamond) | Schoen IWP |

**Figure 1.** Examples of TPMS sheet-based structures generated using the MaSMaker [46] source code. All these structures separate space into two regions of equal volume.

In this paper, MaSMaker v1.0 [46] was used to generate the two TPMS-based structures of interest for studying their behavior as porous media. To characterize the flow through these structures, several cubic unit cells with varying porosity were designed for both structures. Because MaSMaker produces an STL file, the designed structures underwent preprocessing to ensure compatibility with CFD simulation software. This preprocessing involved repairing the triangulated surface in MeshMixer v3.5 [47], followed by a solidification procedure executed in FreeCAD v0.21.1 [48], ultimately allowing their integration into the computational domain of interest.

## 3. Mathematical Model and Numerical Implementation

The computational domain comprises a parallelepiped channel ($60 \times 10 \times 10$ mm$^3$) through which a laminar flow occurs, with a solid TPMS structure (IWP, Schwarz Primitive) immersed within (see Figure 2a). The TPMS unit cells, designed within a volume of $10 \times 10 \times 10$ mm$^3$, are positioned 10 mm from the inlet flow within the parallelepiped domain. Air, characterized by standard constant properties, serves as the working fluid for all reported simulations. As air passes through the structure, the flow and pressure drop along the channel are disturbed due to the presence of the TPMS. This phenomenon is governed by the mass conservation and balance of momentum equations for a laminar, Newtonian, and incompressible flow in a steady state:

$$\nabla \cdot \mathbf{u} = 0, \tag{3}$$

$$(\mathbf{u} \cdot \nabla)\mathbf{u} = -\frac{1}{\rho}\nabla p + \frac{\mu}{\rho}\nabla^2 \mathbf{u}, \tag{4}$$

where $\mathbf{u}$ and $p$ are the velocity and pressure fields, and $\rho$ and $\mu$ are the mass density and dynamic viscosity of the fluid, respectively. The governing equations were discretized using the Finite Volume Method (FVM) implemented in the ANSYS Fluent software [49], where a pressure-based steady solver and a laminar model were selected for solving the governing equations. The Semi-Implicit Method for Pressure-Linked Equations (SIMPLE) algorithm was used for pressure-velocity coupling. The convective terms were discretized using a second-order upwind scheme, while the diffusive terms were approximated using a central differences scheme. The implemented boundary conditions are depicted in Figure 2a. A constant velocity is imposed at the entrance of the channel as the inlet boundary condition, a fixed outlet pressure is applied at the end of the channel, and periodic boundary conditions are enforced in the lateral walls to mitigate their effect, considering that only one unit cell is used in the simulations.

In Figure 2b, an example of the polyhedral surface mesh constructed over the TPMS structure is displayed. A poly-hexahedral volume mesh was constructed, and a central slice of the mesh for the channel with the IWP structure is shown in Figure 2c. This type of mesh predominantly consists of orthogonal cells, resembling a structured mesh

in the bulk of the flow, and offers a good approximation of the geometry near the TPMS structure using a conformal unstructured mesh. To capture the flow inside the pores, appropriate refinements were applied close to the structures. The minimum and maximum cell lengths were defined as 0.02 mm and 0.4 mm, respectively. The constructed meshes achieved acceptable geometrical metrics, with a maximum skewness of 0.72 and a minimum orthogonal quality of 0.23. The convergence of the simulations was ensured by monitoring the resulting residuals, with a value smaller than $10^{-6}$ indicating a converged solution. Additionally, the mass balance at the inlet and outlet of the channel was computed to guarantee mass conservation throughout the system. The Reynolds number based on the width of the channel ($w$) and inlet velocity ($U$) was used for describing the flow regime, $Re = \frac{Uw}{\nu}$.

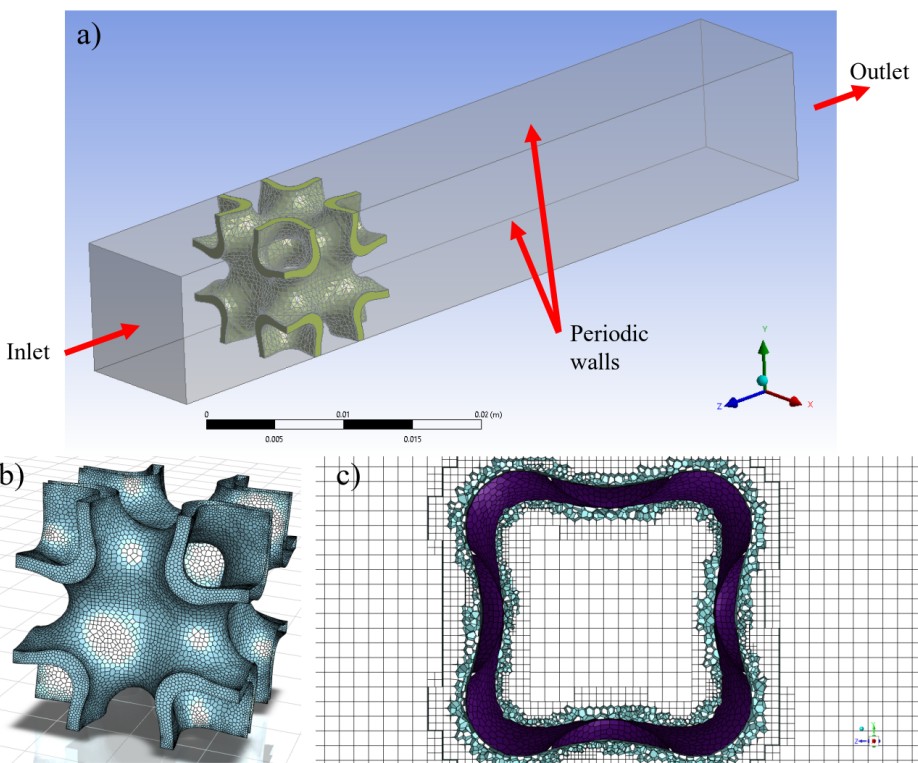

**Figure 2.** (**a**) Physical model and (**b**,**c**) examples of the computational mesh for the IWP structure.

A mesh sensitivity study was conducted to ensure the consistency of the simulations. The analysis was based on the calculation of pressure drop as a function of the computational mesh size for the IWP structure with $t = 0.7$ and Re = 200. First, a coarse mesh with acceptable metrics (orthogonality, skewness, and aspect ratio) was constructed. Then, mesh refinement tools were applied to generate finer grids. Table 1 lists the grids used for the sensitivity analysis along with their corresponding pressure drop results. Notably, grids 3, 4, and 5 exhibited negligible differences in pressure drop. Consequently, computational simulations for all cases were performed using the sizing parameters of grid 4. This allowed for a reduction in computational time while ensuring consistent results for each simulation.

To ensure the physical reliability of the computational mesh, one can assume that when the computational domain is discretized into small control volumes, and the flow is laminar, the variables within each control volume undergo gradual changes. For the case of the velocity change within a mesh cell, this condition is met if the Reynolds number based on the control volume scale is small ($Re_{cv} = zw/\nu < 1$). With this assumption, the $Re_{cv}$ was calculated for each control volume based on its size and velocity, revealing that in around 85% of the total control volumes, it is less than 1.

**Table 1.** Mesh sensitivity analysis results.

| Grid | Cells | $\Delta p/L$ (Pa/m) |
|------|-------|---------------------|
| 1 | 102253 | 389.14 |
| 2 | 258222 | 405.34 |
| 3 | 345876 | 410.10 |
| 4 | 431847 | 409.45 |
| 5 | 760027 | 411.95 |

## 4. Results and Discussion

### 4.1. TPMS Structures Design and Porosity Characterization

TPMS structures were designed in MaSMaker [46], with variations in the isovalue (*t*). As the porosity of the structure depends on *t*, the initial step in modeling such structures as porous mediums involves characterizing the relationship between porosity and the isovalue. For this purpose, Schwarz "P" (also known as Primitive) and IWP structures were designed in MaSMaker for different values of *t*. Using Blender software [50], the volume fraction can be calculated for each structure, and a polynomial fit can be determined to relate the volume fraction ($V_f$) to the isovalue (*t*).

In Figure 3, the results illustrating the relationships between *t* and the volume fraction are presented for both TPMS structures. It is important to mention that each structure has its own topological features. Since our focus is on cases with open and connected porosities allowing fluid passage, the volume fraction range for the two selected structures was restricted to those depicted in Figure 3. The continuous black line represents the best fifth-grade polynomial fit for the resulting relation, calculated using the equation:

$$t = aV_f^5 + bV_f^4 + cV_f^3 + dV_f^2 + eV_f + f. \tag{5}$$

The coefficients of the polynomial fit and their respective coefficients of determination $R^2$ for each TPMS are presented in Table 2. Using this characterization, the porosity ($\phi = 1 - V_f$) of the structures can be directly computed to model them as porous media.

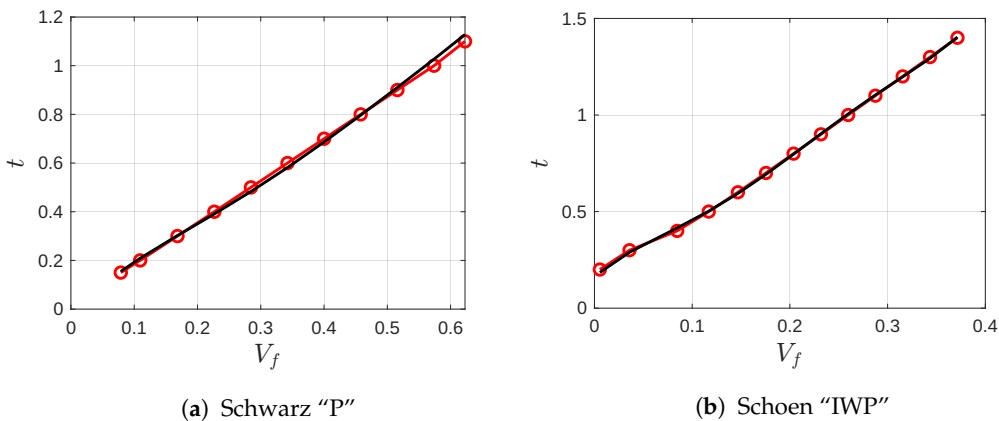

(**a**) Schwarz "P"  (**b**) Schoen "IWP"

**Figure 3.** Isovalue as a function of the volume fraction ($V_f$) for (**a**) Schwarz "P" and (**b**) Schoen IWP structures. The porosity, $\phi$, is related to the volume fraction by $\phi = 1 - V_f$. The red points represent the original data fitted using a fifth-grade polynomial, depicted by the black line.

**Table 2.** Coefficients for the polynomial fit to compute the isovalue as a function of the volume fraction (Equation (5)).

| TPMS | *a* | *b* | *c* | *d* | *e* | *f* | $R^2$ |
|------|-----|-----|-----|-----|-----|-----|-------|
| Primitive | 17.03 | −34.93 | 26.9 | −8.89 | 2.83 | −0.025 | 0.99 |
| IWP | 746.2 | −744.3 | 260.2 | −35.49 | 4.56 | 0.16 | 0.98 |

### 4.2. Mean Flow Characteristics

As an initial exploration, the flow through the TPMS structures was examined. In Figure 4, streamlines depicting the flow are presented for the Schwarz "P" structure with $\phi = 0.83$. It is evident that at low Reynolds numbers (Re = 10), the velocity increases along the central part of the channel, resulting in smooth streamlines that nearly fill the structure channel. At higher Reynolds numbers (Re = 300), recirculations develop at the lateral sides of the structure. The vortices within the Schwarz "P" cell contribute significantly to the pressure drop along the cell, as the flow passing through the structure is confined to the center of the cell.

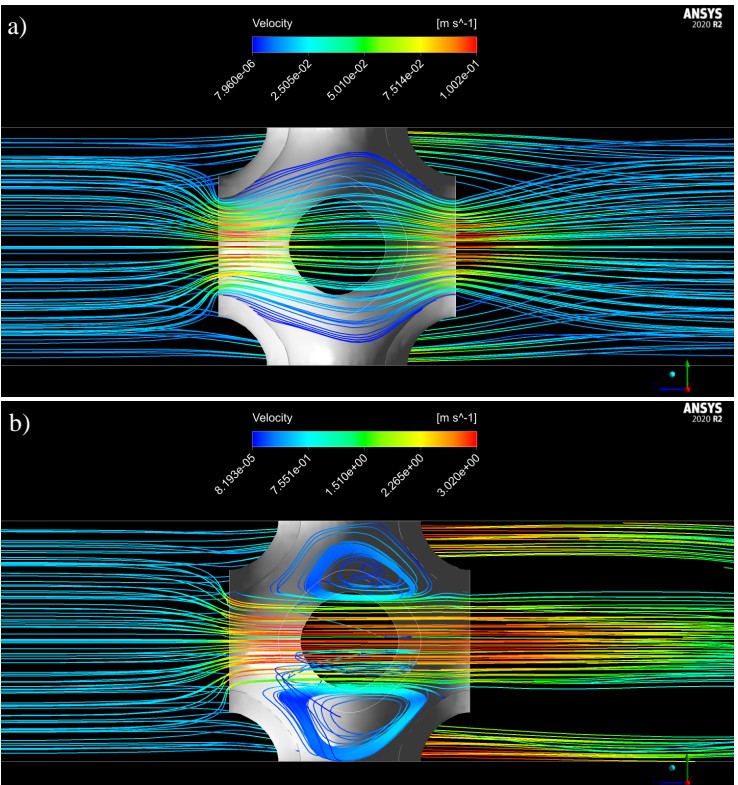

**Figure 4.** Streamlines of the flow through the Schwarz "P" structure ($\phi = 0.83$): (**a**) Re = 10, (**b**) Re = 300.

Streamlines of the flow through the IWP structure are illustrated in Figure 5. As observed, the flow is divided by the channels of the TPMS and smoothly passes through the structure. Specifically, at a low Reynolds number (Re = 10), two symmetrical separated streams can be seen flowing through the central part of the structure. However, at higher Reynolds numbers (Re = 300), the inertial effects of the flow disturb and mix these streams. Under these conditions, the mixing could be advantageous for the application of this type of structure in heat and mass transfer processes.

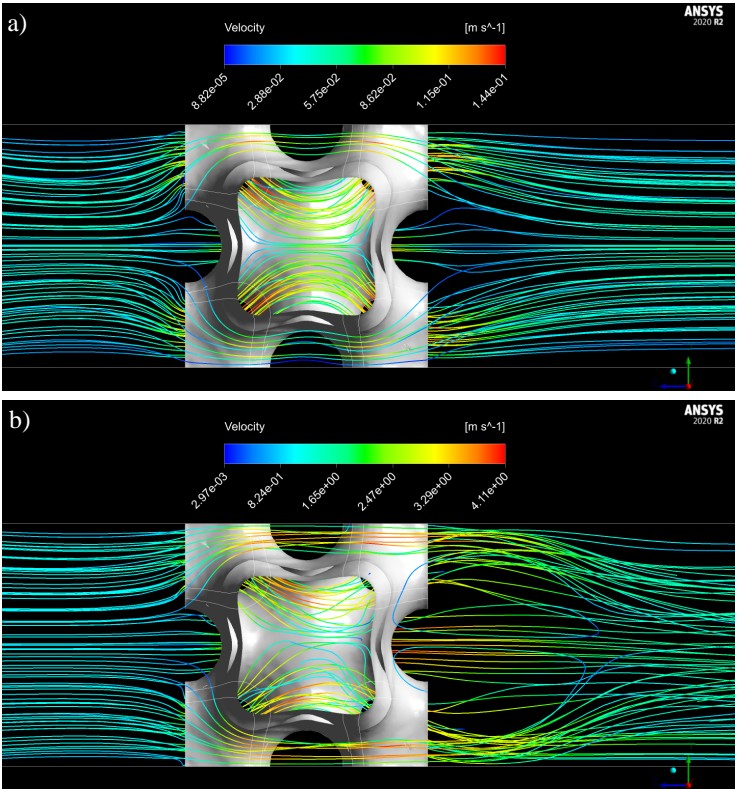

**Figure 5.** Streamlines of the flow through IWP structures ($\phi = 0.79$): (**a**) Re = 10, (**b**) Re = 300.

### 4.3. Modeling the TPMS Structures as Porous Media

The primary objective of this study is to demonstrate how the flow through TPMS structures can be effectively modeled using the Darcy–Forchheimer porous medium model. The aim is to develop relations that enable the computation of pressure drop as a function of porosity and mean flow velocity exclusively. This, in turn, facilitates the straightforward estimation of mean flow characteristics when employing these structures for engineering applications.

The Darcy–Forchheimer model proves to be a common choice for describing fluid flow through porous media, especially when inertial effects are substantial in comparison to diffusive effects. The model amalgamates Darcy's law, asserting that the average flow velocity of a fluid through a porous medium is proportional to the pressure gradient, with an additional term accounting for resistance due to inertia and viscous drag at higher flow rates. Thus, the Darcy–Forchheimer equation takes the form of a two-coefficient parameter model:

$$\frac{\Delta p}{L} = \frac{\mu}{k} U + \frac{c_F \rho}{\sqrt{k}} U^2, \tag{6}$$

where $\Delta p$ is the pressure drop through a porous medium with length $L$, $k$ is the permeability of the medium in the direction of the $u$ component of the velocity, and $c_F$ is the Darcy–Forchheimer coefficient, which is a measure of the resistance of the flow through the porous medium. This resistance considers both viscous drag and inertial effects. The Darcy–Forchheimer coefficient's value depends on various parameters, including porosity, geometric features of the porous medium, and fluid flow conditions. Consequently, a comprehensive understanding of the TPMS structure and its fluid flow characteristics is essential for the accurate application of the Darcy–Forchheimer model.

Computational simulations were systematically conducted, varying the porosity of the structures through the isovalue, $t$, and the inlet stream velocity. In Figure 6, the circle markers represent the pressure drop along the TPMS structures as a function of velocity for different values of $t$. The continuous lines depict the second-grade polynomial fit

applied to each series. Notably, at higher velocities or, equivalently, high *Re*, the pressure drop for Primitive structures is significantly higher (approximately 10 times) than for their IWP counterparts. This discrepancy arises from fluid recirculations within the Primitive structure, as illustrated in Figure 4.

As evident, the second-grade polynomial effectively fits the data obtained from the numerical simulations for all cases. This suggests the potential applicability of the Darcy–Forchheimer model to both TPMS structures.

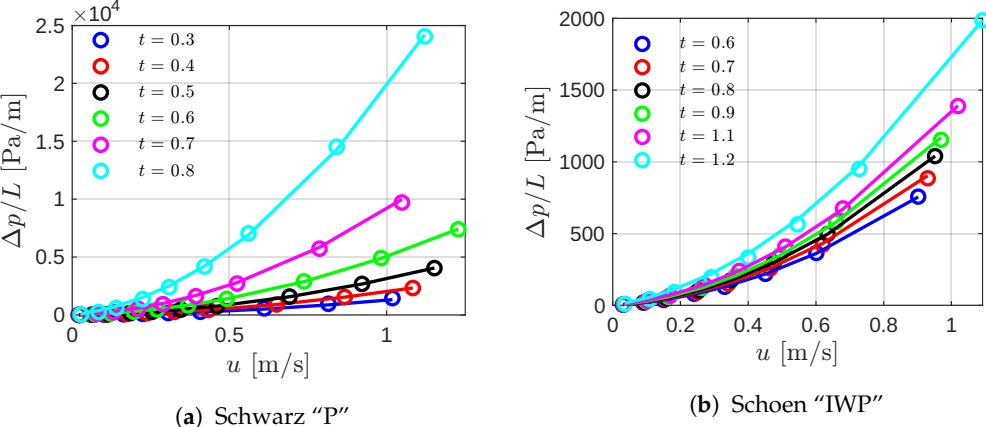

(**a**) Schwarz "P"          (**b**) Schoen "IWP"

**Figure 6.** Pressure drop as a function of the velocity for (**a**) the Primitive and (**b**) IWP structures. Circle markers correspond to the pressure drop computed from the CFD simulations, whereas the continuous lines represent the second-grade polynomial fit.

The next step involves fitting the Darcy–Forchheimer model to the pressure drop predictions obtained from CFD simulations. Subsequently, the permeability and Forchheimer coefficients are computed for each porosity value. In the case of the Primitive structure, the results for permeability and the Darcy–Forchheimer coefficient as functions of porosity are presented in Figure 7. Clearly, there exists a discernible relationship between porosity, permeability, and the Forchheimer coefficient.

Numerous equations have been proposed for different porous media to calculate permeability as a function of porosity [51]. For instance, the Kozeny-Carman equation stands out as one of the most widely recognized relations [52]. Similar to the current study, in [53], the authors examined the pressure drop as a function of velocity for lattice structures. Subsequently, they employed a modified Ergun equation to compute the permeability of the lattices. However, they reported that neither the Ergun equation nor any of its various modifications found in the literature could accurately predict both permeability and the Forchheimer coefficient. In this case, it was also found that the classical correlations are not feasible for predicting the Darcy–Forchheimer model coefficients to compute the pressure drop through these kinds of structures. Nevertheless, other models have been proposed for relating the permeability with porosity [51], for instance, power law regressions. Following this idea, for the Primitive structure, power law relations were computed for both permeability and Forchheimer coefficient according to:

$$k(\phi) = a\phi^b; \qquad c_F(\phi) = d\phi^e, \tag{7}$$

where *a*, *b*, *c* and *d* are constant coefficients used to fit the power law.

The power law fits for permeability, and Forchheimer coefficients are plotted as continuous lines in Figure 7a,b, respectively. The resulting coefficients of the power laws are reported in Table 3. Notably, the power laws fit very well with the values of *k* and $c_F$ computed from the numerical simulations. The correlation coefficient ($R^2$) for permeability and Forchheimer coefficients power law fits are 0.982 and 0.994, respectively.

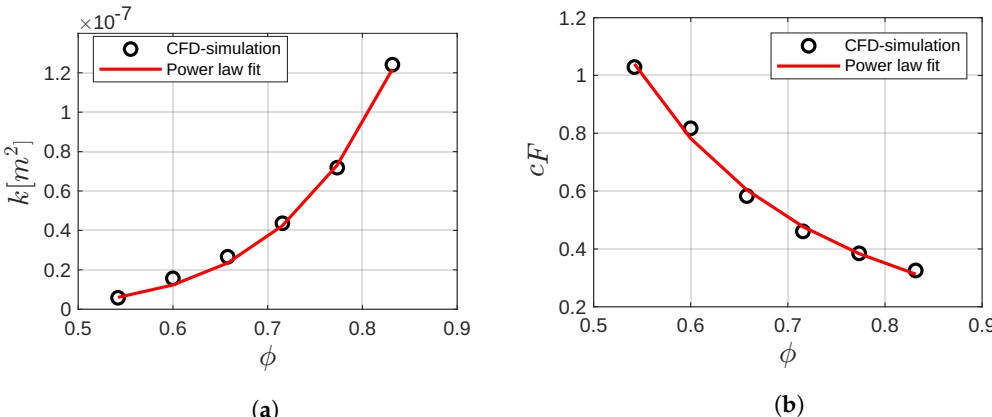

(**a**)         (**b**)

**Figure 7.** (**a**) Permeability and (**b**) Forchheimer coefficient as a function of the porosity for the Primitive structure.

**Table 3.** Coefficients for fitting the power laws for $k$ and $c_F$.

| | $k$ | | $C_F$ | |
|---|---|---|---|---|
| TPMS | $a$ | $b$ | $d$ | $e$ |
| Primitive | $4.439 \times 10^{-7}$ | 7.015 | 0.1865 | $-2.806$ |
| | $d$ | $e$ | $f$ | $C_F$ |
| IWP | $1.005 \times 10^{-6}$ | 0.4505 | $-7.971 \times 10^{-7}$ | $\approx 0.25$ |

For the IWP structure, it was found that it is also possible to fit a power law for calculating the permeability as a function of the porosity. The results are shown in Figure 8. In this particular case, due to the geometric features of the IWP structure, the power law added an extra constant term:

$$k(\phi) = d\phi^e + f. \tag{8}$$

The term $f$ is necessary since, at lower porosities, the IWP structure design features isolated void spaces through which the fluid cannot flow. Consequently, permeability under such circumstances approaches zero. In Figure 8, the permeability as a function of porosity is presented, with circle markers indicating results obtained from CFD simulations. The continuous red line represents the power law proposed in Equation (8). Notably, the power law aligns well with the simulation results, yielding a coefficient of determination $R^2 = 0.98$. Regarding the Forchheimer coefficient, it was determined that within the porosity range analyzed in this study, a constant value of 0.25 suffices for predicting pressure drop using the Darcy–Forchheimer model.

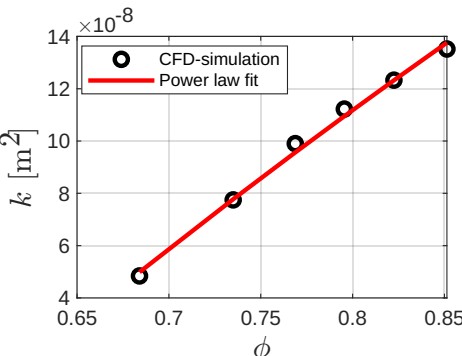

**Figure 8.** Permeability as a function of the porosity for the IWP structure.

Finally, the Darcy–Forchheimer model was used to compute the pressure drop as a function of the stream velocity varying the isovalue, for both structures. The permeability

and Forchheimer coefficients were directly calculated by the proposed power law equations (Equations (7) and (8)). In Figure 9, the pressure drop computed by the porous media model is compared with the results from the CFD simulations for both the Primitive and IWP structures. In this case, the circle markers are the CFD results, and the continuous lines are the prediction calculated by the Darcy–Forchheimer model. Notably, the model exhibits good agreement with the simulations within the range of porosity and flow velocity analyzed. For the Primitive and IWP structures, the mean error of the computed pressure drop by Equation (7) relative to CFD calculations is approximately 11% and 5%, respectively.

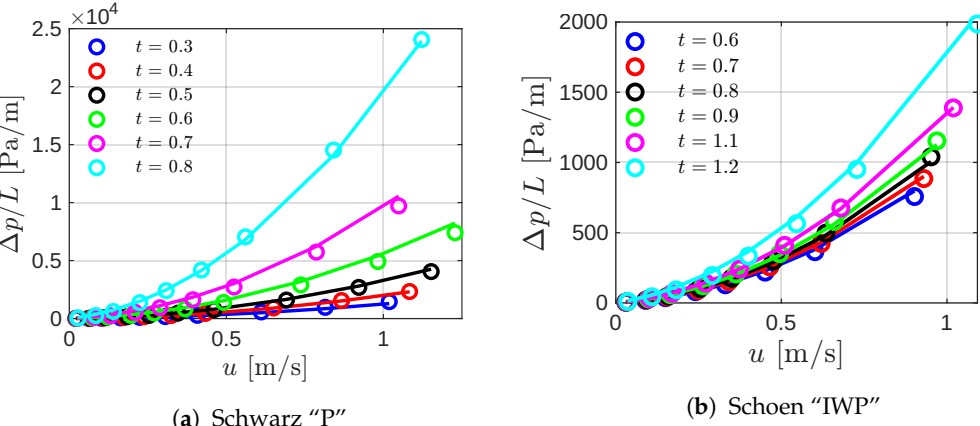

(**a**) Schwarz "P"             (**b**) Schoen "IWP"

**Figure 9.** Pressure drop as a function of the velocity for (**a**) the Primitive and (**b**) IWP structures. Circle markers are the pressure drop computed from the CFD simulations, whereas the continuous lines are the pressure drop calculated using the Darcy–Forchheimer model coupled with the proposed power law relations for permeability and Forchheimer coefficient.

## 5. Conclusions

In this paper, the fluid flow analysis through two different TPMS structures was presented. The studied structures were designed, preprocessed, and geometrically characterized to establish a relationship between the volume fraction and the isovalue. It was found that a fifth-grade polynomial adequately fits both variables for both structures. Computational simulations were conducted to characterize the flow inside the Primitive and IWP structures under laminar regimes. The main features of the mean flow were discussed for both structures, revealing that the Primitive structure promotes the occurrence of fluid recirculations inside the cell at high Reynolds numbers. Pressure drop as a function of the stream velocity was also computed for different structure porosities. The Primitive structure exhibits a larger pressure drop, approximately ten times larger than the IWP structure, attributed to the earlier mentioned fluid recirculations. Subsequently, it was demonstrated that the flow inside both structures can be modeled by the Darcy–Forchheimer equation for a porous medium. A key result of this investigation is the proposal of power laws to calculate the permeability and Forchheimer coefficient as functions of the porosity for the analyzed structures. Power law equations have been widely employed to describe the permeability of porous media [52]. The proposed relations enable direct computation of the pressure drop along the structures using a porous media model. The mean errors between the pressure drop computed from the Darcy–Forchheimer model and that from the CFD simulations were 11% and 5% for the Primitive and IWP structures, respectively. These results can be utilized in the early stages of the design process for components in fluid flow applications based on TPMS since the pressure drop can be easily calculated only by establishing the porosity of the structure and the mean flow velocity.

Given that the design of heat exchangers requires knowledge of both the pressure drop and the convective heat transfer coefficient in the system, this study contributes to the design process for these devices. The presented results can serve as a basis for the design of heat transfer applications, provided that correlations for the Nusselt number (or,

equivalently, the convective heat transfer coefficient) are known for the TPMS structures considered, which is still an open problem.

**Author Contributions:** Conceptualization, S.P.; methodology, S.P.; software, S.P.; validation, S.P. and J.P.-B.; formal analysis, S.P.; investigation, S.P., A.G.-O. and J.P.-B.; writing—original draft preparation, S.P.; writing—review and editing, A.G.-O. and J.P.-B.; visualization, S.P. and J.P.-B.; supervision, A.G.-O. and J.P.-B.; project administration, S.P. and J.P.-B. All authors have read and agreed to the published version of the manuscript.

**Funding:** Publication of this research was funded through CONAHCYT's project No. 322615—"F003 Desarrollo de tecnología incremental y disruptiva en sistemas de enfriamiento 2023–2024".

**Data Availability Statement:** The data presented in this study are available from the corresponding author upon reasonable request.

**Acknowledgments:** The authors acknowledge support from CONAHCYT's project No. 321159 "Laboratorio Nacional de Investigación en Tecnologías del Frío". S. Piedra, A. Gómez-Ortega, and J. Pérez-Barrera thank the "Investigadoras e Investigadores por México" program from CONAHCYT. Support from the National Cooperative for Additive Manufacturing (CONMAD-CIDESI) is acknowledged.

**Conflicts of Interest:** The authors declare no conflict of interest. The funding sponsors had no role in the design of the study; in the collection, analyses, or interpretation of data; in the writing of the manuscript, and in the decision to publish the results

## Abbreviations

The following abbreviations are used in this manuscript:

| | |
|---|---|
| AM | Additive Manufacturing |
| CFD | Computational Fluid Dynamics |
| DF | Darcy–Forchheimer |
| FVM | Finite Volume Method |
| SIMPLE | Semi-Implicit Method for Pressure-Linked Equations |
| TPMS | Triply Periodic Minimal Surface |

## Nomenclature

| | |
|---|---|
| $a$ | Unit cell size (m) |
| $c_F$ | Forchheimer coefficient |
| $k$ | Permeability ($m^2$) |
| $\mu$ | Dynamic viscosity (kg m$^{-1}$ s$^{-1}$) |
| $\nu$ | Kinematic viscosity, $\mu/\rho$ (m$^2$ s$^{-1}$) |
| $p$ | Pressure field (N m$^{-2}$) |
| $\phi$ | TPMS porosity, $1 - V_f$ |
| $Re$ | Reynolds number, $Uw/\nu$ |
| $\rho$ | Mass density (kg m$^{-3}$) |
| $t$ | Isovalue of the TPMS |
| **u** | Vector velocity field (m s$^{-1}$) |
| $U$ | Inlet flow velocity (m s$^{-1}$) |
| $V_f$ | TPMS Solid volume fraction |
| $w$ | Width of the channel (m) |

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
