# Peer review of "Prediction of Flow Properties of Porous Triply Periodic Minimal Surface (TPMS) Structures"

_fluids, doi:10.3390/fluids8120312_

Round 1

Reviewer 1 Report

Comments and Suggestions for Authors

The present study, laminar flow through Triply Periodic Minimal Surface (TPMS) structures was numerically analyzed using CFD simulations. Porosity of the structures was characterized by computing the volume fraction as a function of the structure isovalue, then performing a polynomial fit. CFD simulations were implemented for computing the pressure drop as a function of the inlet velocity for the flow through those structures using different volume fractions, and then a Darcy-Forchheimer model was fitted to those results in order to calculate the inertial coefficient and permeability as functions of the porosity.

The topic fits the scope of the journal Fluids.

This paper is well organized. The research background is solid. The results of this manuscript are well presented and organized. The introduction is written with an overview and cited literature based on the issue. Methods are properly interpreted and described.

The following issues may be considered and revised:

1.       Research status on heat exchangers integrating these TPMS structures should be given in introduction part. Would those TPMS structures used in some real applications? The introduction part focused on the model development etc. for a major part.

2.       It is difficult to validate this CFD model with an experiment results. Could the model be validated against some other experimental results in other publications?

3.       The descriptions on CFD model should be implemented, for example which laminar model?

4.       Show the full name of abbreviations when it appears first time, many abbreviations were not given although it was common sense. Please double check.

Reviewer 2 Report

Comments and Suggestions for Authors

This paper deals with the geometrically complex structure in prediction of flows properties of porous triply periodic minimal surface structure, which presents the numerical simulation by using CFD technique. The results seems correct, which is well written. Hence, I am glad to recommend this paper to be accept for publication except some revisions:

(1) The CFD should use the complete name in abstract.

(2) The innovation should be pointed in introduction.

(3) The relation between the problem and porous flow has not been stated.

(4) The references should add some results related to mathematical theory about this model.

Comments on the Quality of English Language

The anstract should be improved, which seems too verbose, and the highlights have not been presented.

Reviewer 3 Report

Comments and Suggestions for Authors

ln.128

I believe that "Schwarz primitive" is a special type of primitive function associated with a family of conformal mappings...

Please define IWP acronym...

Is it related with some kind of "Schwarz primitive" also?

Whats the meaning of IWP acronym?

Dose it refers to "Schoen I-graph-wrapped package"?

-------------

ln.209-...

"best fifth-grade polynomial fit"

> pleas specify what are your considered criteria for "best polynomial fit"?

> what about statistical model choice strategy and statistical significance of model parameters? 

-------------

Table 2. Coefficients

> since you have estimated model coefficients...  What about respective uncertainties and statistical significance?

> parameter uncertainties should be estimated also! (central estimates are meaningless without respective uncertainties)

-------------

Figure 3

> Assuming that red points correspond to initial data to be fitted...

I don't believe both situations (left and wright) represents the same experimental situation!

> please use SAME SCALE in both plots (left and wright) and correct this images

-------------

eq.(6)

> please add the meaning of "L" (path length?)

-------------

Table 3. Coefficients...

> same comments as in "Table 2" -> parameter uncertainties should be estimated also!

I now that you are now working with "Non-linear models" (cross-dependent sensitivity coefficents) with "fallible" uncertianties

that have to be evaluated by computational simulation methods (e.g. Monte-Carlo or ressampling methods)

-------------
